# Ethylene Inhibits Anthocyanin Biosynthesis by Repressing the R2R3-MYB Regulator *SlAN2-like* in Tomato

**DOI:** 10.3390/ijms23147648

**Published:** 2022-07-11

**Authors:** Yulian Xu, Xiaoxi Liu, Yinggemei Huang, Zhilei Xia, Zilin Lian, Lijuan Qian, Shuangshuang Yan, Bihao Cao, Zhengkun Qiu

**Affiliations:** 1College of Horticulture, South China Agricultural University, Key Laboratory of Biology and Genetic Improvement of Horticultural Crops (South China), Ministry of Agriculture and Rural Affairs/Guangdong Vegetable Engineering and Technology Research Center, Guangzhou 510642, China; 18826074460@163.com (Y.X.); ygmhuang@163.com (Y.H.); zhilei18339431441@163.com (Z.X.); zlinlian@163.com (Z.L.); Qlj525009@163.com (L.Q.); ssyan@scau.edu.cn (S.Y.); 2Guangdong Key Laboratory of New Technology Research of Vegetable, Vegetable Research Institute, Guangdong Academy of Agricultural Sciences, Guangzhou 510640, China; liuxxabc@163.com

**Keywords:** tomato, anthocyanins, ethylene, *SlAN2-like*, transcriptome

## Abstract

Fruit ripening is usually accompanied by anthocyanin accumulation. Ethylene is key in ripening-induced anthocyanin production in many fruits. However, the effects of fruit ripening and ethylene on anthocyanin biosynthesis in purple tomato fruits are unclear. This study shows that bagged fruits of the purple tomato cultivar ‘Indigo Rose’ failed to produce anthocyanins at the red ripening stage after bag removal. In contrast, the bagged immature fruits accumulated a significant amount of anthocyanins after removing the bags. The transcriptomic analyses between immature and red ripening fruit before and after bag removal revealed that anthocyanin-related genes, including the key positive R2R3-MYB regulator SlAN2-like, were repressed in the red ripening fruit. The 86 identified transcription factors, including 13 AP2/ERF, 7 bZIP, 8 bHLH and 6 MYB, showed significantly different expressions between immature and red ripening fruits. Moreover, subjecting bagged immature fruits to exogenous ethylene treatment significantly inhibited anthocyanin accumulation and the expression of anthocyanin-related genes, including the anthocyanin structure genes and *SlAN2-like*. Thus, ethylene inhibits anthocyanin biosynthesis by repressing the transcription of *SlAN2-like* and other anthocyanin-related genes. These findings provide new insights into anthocyanin regulation in purple tomato fruit.

## 1. Introduction

Anthocyanins are a class of flavonoids widely distributed in plant leaves, flowers, and fruits [1]. Anthocyanins confer different plant pigments, including orange, brown, red, blue and purple, and protect plants against various biotic and abiotic stresses, including those caused by insects, phytopathogens, drought, UV irradiation and low temperatures [2,3,4,5]. Notably, several studies of human subjects and animal systems have demonstrated that anthocyanins benefit humans through their antioxidant activity and ability to induce protective enzymes [6,7,8].

The genes encoding anthocyanin biosynthesis enzymes are characterized into two classes, early biosynthetic genes (EBGs) and late biosynthetic genes (LBGs) [9,10,11]. EBGs, including *phenylalanine ammonia lyase* (*PAL*), *4-coumaryl:CoA ligase* (*4CL*), *chalcone synthase* (*CHS*), *chalcone isomerase* (*CHI*) and *flavanone 3-hydroxylase* (*F3H*), produce common flavonoid precursors. In contrast, LBGs, including *flavonoid 3′5′-hydroxylase* (*F3′5′H*), *dihydroflavonol 4-reductase* (*DFR*), *anthocyanidin synthase* (*ANS*), *flavonol-3-glucosyltransferase* (*3GT*), *rhamnosyl transferase* (*RT*), and *anthocyanin acyltransferase* (*AAC*), specifically regulate anthocyanin and proanthocyanin accumulation. Anthocyanins are synthesized in the cytosol and finally transported into cell vacuoles by the putative anthocyanin transporter (PAT) or glutathione-S-transferase (GST) [12]. A conserved MBW ternary complex comprising a WD-repeat protein and R2R3-MYB and bHLH transcription factors (TFs), regulates the transcriptional biosynthesis of anthocyanins [13,14]. However, R2R3-MYB TFs are the key regulators that spatiotemporally control the expression of anthocyanin structural genes [15,16]. R2R3-MYB TFs also play a critical role in ethylene-induced anthocyanin production [17].

Ethylene is an important gaseous phytohormone involved in plant development [18]. Additionally, many studies have revealed that ethylene induces anthocyanin biosynthesis. For example, exogenous ethylene increases anthocyanin accumulation in grape berries by up-regulating the expression of anthocyanin structural genes [19]. Similar results are reported in plum [20,21], apple [22], and strawberry [23]. In apple, two ethylene response factors, MdERF1B and MdERF3, are key positive regulators of anthocyanin biosynthesis in response to ethylene signals [24].

Recently, MdEIL1, a critical component of the ethylene signaling pathway, was characterized to form a regulatory module with MdMYB17 and MdMYB1 for the fine modulation of ethylene-regulated anthocyanin production in apple fruits [17]. However, ethylene can negatively control anthocyanin accumulation in some plants. For example, exogenous ethylene inhibits anthocyanin biosynthesis in sorghum [25], Arabidopsis [26], and pear [27]. In Arabidopsis, ethylene inhibits sugar-induced anthocyanin accumulation by suppressing the transcription of positive regulators, such as *GL3*, *TT8*, and *PAP1*. However, it increases the transcription of negative regulators, such as *MYBL2* [26,28]. Recently, with the co-expression network and Mfuzz analyses, an ethylene response factor (ERF), *PpERF105* was identified [27]. *PpERF105*, induced by ethylene, activates the expression of the R2R3-MYB repressor *PpMYB140* to inhibit anthocyanin production in pear [29].

Cultivated tomatoes, one of the most consumed vegetables globally, produce anthocyanins in vegetative tissues but not fruits [30]. The high anthocyanin levels in tomato fruit improve the nutrients of tomatoes, doubling the shelf life of tomato fruit by delaying overripening and reducing susceptibility to grey mold [31,32,33,34]. For ten years now, efforts have focused on increasing the anthocyanin content of tomato fruits [35,36,37,38,39]. Myers first bred a purple tomato cultivar ‘Indigo Rose’ by introgressing the *Anthocyanin fruit* (*Aft*) and *atroviolacea* (*atv*) loci into domesticated tomatoes from different wild tomatoes [40]. The total anthocyanin content in the ‘Indigo Rose’ peel reached 200 mg/100 g of fresh weight [41,42]. The main anthocyanin in the peel of ‘Indigo Rose’ is petunidin-3-(trans-p-coumaroyl)-rutinoside-5-glucoside [43].

*Aft* encodes an R2R3-MYB TF, *SlAN2-like*, that positively regulates anthocyanin biosynthesis in tomato fruits [39,42,44]. A protein interaction analysis indicated that SlAN2-like interacts with the bHLH factor SlJAF13 and the WD-repeat protein SlAN11 to form an MBW (SlAN2-like-SlJAF13-SlAN11) complex [42,45]. This complex regulates *SlAN1* expression and anthocyanin biosynthesis genes (EBGs and LBGs), resulting in fully purple-skinned tomatoes [42]. *SlAN1* is a bHLH factor crucial for anthocyanin biosynthesis in tomatoes [3]. *ATV* encodes an R3-MYB factor, *SlMYBATV*, which negatively controls anthocyanin accumulation in tomato fruits [39,41,42,46]. In purple-spotted tomato plants, the SlAN2-like-SlJAF13-SlAN11 MBW complex activates *SlMYBATV*, which competes with SlAN2-like for SlJAF13, inhibiting the transcription of *SlAN1* and anthocyanin biosynthesis genes [42].

Tomato is a classical climacteric fruit, exhibiting a rapid rise in respiration and a burst of ethylene production during ripening initiation [47]. However, the effects of the ripening stage and ethylene on anthocyanin biosynthesis in anthocyanin-rich tomato fruits are still poorly understood. In this study, the bagged fruit of red ripened ‘Indigo Rose’ fruits fail to accumulate anthocyanins after exposure to light. Combining RNA-seq analysis, exogenous ethylene treatment, and relative expression analysis, we confirm that ethylene inhibits anthocyanin biosynthesis by repressing *SlAN2-like* expression in tomatoes. These results provide new insights into the regulatory effects of ethylene on anthocyanin biosynthesis in fruits.

## 2. Results

### 2.1. Evaluating the Color and Anthocyanin Content of Bagged Fruits after Removing Bags

Light is considered indispensable for anthocyanin induction [48]. Our previous studies indicated that ‘Indigo Rose’ bears purple–black fruits with high anthocyanin content in their peel at and after the mature green stage [41]. Here, we showed that bagged ‘Indigo Rose’ fruits from the immature or red ripening stages were white or red (Figure 1A). Moreover, fruits from both stages had undetectable anthocyanin content in their peels (Figure 1B). After removing bags, the immature fruits gradually turned purple, accompanying the increasing anthocyanin content, while fruits from the red ripening stage remained red, with a few expressing increased anthocyanin levels (Figure 1). These results confirmed that light is necessary for anthocyanin biosynthesis in purple tomatoes. These things considered, we speculate that some unknown underlying factors repress anthocyanin biosynthesis during the red ripening stage.

### 2.2. Transcriptome Analysis of the Fruit Peel before and after Bag Removal

RNA-seq analyses were performed on the peels of fruits from the immature (IM) or red ripening (RR) stages before (0 days, 0D) and after (2D) bag removal to uncover the molecular mechanism of anthocyanin biosynthesis. In total, 243.6 million of the obtained clean reads were unique and mapped to the tomato genome (SL4.1, Appendix A). The unique mapped reads were used for FPKM calculation and for identifying differentially expressed genes (DEGs). Principal component analysis (PCA) showed that the three biological replicates of each sample were clustered together, indicating the high repeatability of the experiments (Appendix A).

At *p* < 0.01, the pairwise comparison of the four treatments identified 3302 DEGs (Figure 2A and Appendix A). The DEGs were clustered into 11 groups with distinct expression patterns (Figure 2A and Appendix A). The genes from cluster 6 were up-regulated at 0D and 2D, with higher expression levels in the peels of immature fruits after bag removal than in the red ripening stage (Figure 2B). This is consistent with the anthocyanin content in the fruits (Figure 1B); thus, it was selected for further analyses. GO enrichment analysis of the biological processes showed that the genes from cluster 6 enriched the ‘response to light’, ‘chloroplast organization’ and ‘chlorophyll biosynthetic process’ pathways. Coinciding with the anthocyanin content, the genes from cluster 6 also enriched the ‘flavonoid metabolic process’ (FDR = 0.000188, Figure 2C).

Most anthocyanin structural genes, including EBGs (*SlPAL*, *SlC4H*, *Sl4CL*, *SlCHI*, and *SlF3H*) and LBGs (*SlDFR*, *SlANS*, *Sl3-GT*, *SlRT1*, *SlRT2*, *SlAAC*, and *SlGST*), were in cluster 6, showing significantly higher expression in the peels of immature fruits after bag removal (IM-2D, Figure 2D). Previous studies revealed that the R2R3-MYB factor *SlAN2* regulates anthocyanin biosynthesis in vegetative tissues [49]. In contrast, its paralogue, *SlAN2-like*, only regulates anthocyanin accumulation in the peels of tomato fruits [42]. *SlAN2* and *SlAN2-like* were both highly expressed after bag removal but barely expressed in bagged fruits (OD) at both immature (IM) and red ripening (RR) stages (Figure 2D). Moreover, *SlAN2* (38.75-fold) and *SlAN2-like* (4.60-fold) expressions were much higher in immature fruit peels than in the red ripening stage, correlating with the anthocyanin content (Figure 2D and Appendix A). The FPKM of *SlAN2-like* (596.22) was higher than *SlAN2* (FPKM = 35.11) in the IM-2D samples (Appendix A). Two bHLH transcription factors, *SlAN1* and *SlJAF13*, showed significantly higher expression levels in IM-2D than in RR-2D, except for R2R3-MYB regulators (Figure 2D). In addition, the flavonol-related genes, *SlF3′H* and *SlMYB12*, were highly expressed in IM-2D (Figure 2D).

### 2.3. Anthocyanin-Related Genes Showing Different Expression Patterns between Immature and Red Ripening Fruits after Bag Removal

The qRT-PCR experiment was performed to analyze the expression pattern of anthocyanin-related genes in immature and red ripening fruits before and eight days after bag removal. The structural genes, including EBGs (*4CL*, *CHI*, and *F3H*) (Figure 3A) and LBGs (*F3′5′H*, *DFR*, and *ANS*) (Figure 3B), rapidly increased in immature fruit peels after bag removal, with most reaching their peak five days after bag removal. A similar pattern of structural gene expression was detected in fruits in the red ripening stage after removing bags (Figure 3A,B). However, the expression was significantly lower in fruits in the red ripening stage than in immature fruits after exposure to light (Figure 3A,B).

As with the structure genes, the key anthocyanin activators, *SlAN2-like* and *SlAN1*, increased quickly in immature fruits after bag removal (Figure 3C), but differed from the red ripening fruits. Similarly, *SlAN2-like* rapidly increased in red ripening fruits after bag removal, but the expression was significantly lower than in the immature fruits (Figure 3C). *SlAN1* was barely expressed in the red ripening fruits, even after bag removal (Figure 3C). *SlHY5*, a light-induced bZIP factor, activated anthocyanin biosynthesis [50]. *SlHY5* was more highly expressed in red ripening than in immature fruits at 0D and 8D but was insignificantly different at 2D and 5D (Figure 3C).

### 2.4. Transcriptome Analysis of the DEGs between Immature and Red Ripening Fruits

We previously speculated that some factors in red ripening fruits repress anthocyanin biosynthesis. Hence, the differentially expressed genes between immature and red ripening fruits were analyzed to characterize the factors repressing anthocyanin biosynthesis. Finally, five clusters (1, 4, 8, 9, and 11) that could be classified into two larger groups were used for further analysis. Group 1 included clusters 4, 9, and 11, containing genes with lower expression levels in immature than in red ripening fruits (Figure 4A). In contrast, group 2 included clusters 1 and 8, which contained genes with higher expression in immature than in red ripening fruits (Figure 4B). GO enrichment analysis (biological process) showed that group 1 genes enriched ‘chloroplast organization’, ‘response to light stimulus’, and the ‘carotenoid biosynthetic process’ (Figure 4C). Group 2 genes enriched the ‘monocarboxylic acid biosynthetic process’, ‘cytoskeleton organization’, and ‘cuticle development’ (Figure 4D).

Group 1 genes contained 42 transcription factors, namely, 6 AP2/ERF (Solyc03g044300.3, Solyc04g009450.1, Solyc11g010710.2, Solyc07g054220.1, Solyc10g076370.3, and Solyc04g071770.3), 4 bZIP (Solyc01g109880.3, Solyc09g009760.1, Solyc01g100460.3, and Solyc05g050220.3), 4 NAC (Solyc07g063420.3, Solyc02g077610.3, Solyc05g055470.4, and Solyc11g008010.2), 2 bHLH (Solyc01g109700.3 and Solyc08g076930.1), and 1 MYB (Solyc11g073120.2) and WRKY (Solyc08g006320.5) (Appendix A). However, group 2 had 44 transcription factors, including 3 AP2/ERF (Solyc02g064960.3, Solyc04g014530.1, and Solyc05g051200.1), 3 bZIP (Solyc04g011670.3, Solyc04g072460.3, and Solyc12g010800.2), 6 bHLH (Solyc06g008030.3, Solyc09g097870.4, Solyc10g079070.2, Solyc06g051260.4, Solyc07g043580.4, and Solyc08g081140.4), 5 MYB (Solyc02g088190.5, Solyc10g081320.1, Solyc01g096700.4, Solyc01g095030.3, and Solyc12g044610.2) and 1 WRKY (Solyc12g014610.2) (Appendix A).

### 2.5. Exogenous Ethylene Repressed Anthocyanin Accumulation and Anthocyanin-Related Genes

Ethylene regulates anthocyanin biosynthesis by moderating the activities of R2R3-MYB in many plants [24,29]. Thus, four ethylene-related genes, *ACS2*, *Solyc08g081535.1* (ACO-family), *Solyc02g036350.3* (ACO-family), and *ACO3*, annotated as ethylene metabolism genes, were more highly expressed in the red ripening than the immature fruits (Figure 5A). In addition, three genes of the ethylene signaling pathway, *GRL2*, *NR*/*ETR3*, and *EIL3*, were more highly expressed in the red ripening compared to the immature fruits (Figure 5A).

The bagged fruits (before maturation) were treated with ethephon (ETH) to evaluate the effects of ethylene on the anthocyanin biosynthesis of the tomato fruit. As a result, the mock-treated fruits had higher purple-pigment intensities than the ETH-treated fruits (Figure 5B). Therefore, the mock-treated fruits had significantly higher anthocyanin contents than the ETH-treated fruits (Figure 5C). These results indicate that ethylene repressed anthocyanin biosynthesis in purple tomato fruits.

The qRT-PCR relative expression analysis of anthocyanin EBGs (*4CL* and *CHI*) showed a non-significant difference between the ETH-treated and mock fruits, but *F3H* was more highly expressed in mock than in ETH-treated fruits at 5D (Figure 5D). Unlike EBGs, all the LBGs, including *F3′5′H*, *DFR*, and *ANS*, were more repressed in the ETH-treated compared to the mock fruit (Figure 5E). Similarly, *SlAN2-like* and *SlAN1* genes were all more repressed in the ETH-treated fruits relative to the mock (Figure 5F). However, *SlHY5* was up-regulated more in ETH-treated fruits than in the mock fruits (Figure 5F).

## 3. Discussion

Tomato is a climacteric fruit with undetectably low ethylene contents before the start of fruit ripening [46]. Previously, high anthocyanin contents were detected in immature and mature green fruits of ‘Indigo Rose’ [43,50], indicating that ethylene is not essential for anthocyanin biosynthesis in tomato fruits. Thus, the bagged immature fruits gradually turned purple with anthocyanin accumulation (Figure 1). At the red ripening stage, the bagged fruits did not produce anthocyanins after bag removal (Figure 1). This response suggests that anthocyanin biosynthesis was not induced or was highly repressed in the red ripening fruit after exposure to natural light.

Moreover, the combined transcriptome and relative expression data showed that anthocyanin-related genes, including the structure and regulatory genes, were more repressed in the red ripening fruits compared to the immature fruits (Figure 2D and Figure 3). *SlHY5* was previously characterized as a key anthocyanin regulator in light-exposed tomatoes [50]. The expression of *SlHY5* was indifferent between immature and red ripening fruits after bag removal (2D and 5D; Figure 3C). This pattern suggests that other factors/genes inhibit the transcription of anthocyanin-related genes in the red ripening fruits after bag removal.

In many fruit tree crops, fruit ripening is usually accompanied by anthocyanin accumulation, and ethylene is considered an anthocyanin activator in plants. Several studies have confirmed that ethylene induces red/purple coloration by activating anthocyanin production in apple, plum, grape, and strawberry [19,20,21,51]. However, ethylene inhibited anthocyanin accumulation in Arabidopsis [26], pear [27], and immature fruits of *Fragaria chiloensis* [52]. Thus, the effect of ethylene on anthocyanin biosynthesis varies significantly among species and needs further exploration. The ethylene content in tomato fruits was highest at the onset of ripening (breaking stage) and sharply decreased during fruit ripening. However, it remained higher in the red ripening fruits than in the immature or mature green fruits [47]. Indeed, the ACS family of ethylene biosynthetic genes (*ACS2* and *Solyc08g081535.1*) and the SlEBF family of ethylene signaling genes (*ETR3*, *EIL3* and *Solyc07g008250.3*) were more highly expressed in the red ripening than in the immature fruits (Figure 5A). Thus, we hypothesized that the ethylene in the bagged red ripening fruits probably effects anthocyanin production.

The ETH-treated fruits had lighter purple pigmentation than the mock fruits (Figure 5B). Consistent with the above, the visual observation showed lower anthocyanin content in the ETH-treated fruits than in the mock fruits (Figure 5C). Similarly, ethylene inhibited light-induced anthocyanin biosynthesis in the red pear fruits [27]. Moreover, the transcriptome data showed that anthocyanin- and flavone-related genes (*PpANS*, *PpUFGT2*, *PpMYB10*, and *PpMYB114*) were positively and highly correlated with anthocyanin content, meaning that these genes were repressed in exogenous ethylene-treated pear fruits [27]. In tomatoes, the anthocyanin latter biosynthesis (*F3′5′H*, *DFR*, and *ANS*) and the key positive regulatory genes (*SlAN2-like* and *SlAN1*) were more down-regulated in the ETH-treated fruits than in the mock fruits (Figure 5D–F). In addition, several transcription factors, including ERF, MYB, and bHLH, were identified from the transcriptome analysis and were highly correlated with the anthocyanin content (Appendix A). Nevertheless, their function in anthocyanin regulation in tomato fruit requires further studies. These results imply that ethylene inhibits anthocyanin production by repressing *SlAN2-like* expression. However, the regulatory efforts of ethylene on *SlAN2-like* expression require further exploration.

In summary, in this study, we investigated the effects of fruit ripening and ethylene on anthocyanin biosynthesis in purple tomato fruits. We found that bagged fruits of the purple tomato cultivar ‘Indigo Rose’ failed to produce anthocyanins at the red ripening stage after bag removal. In contrast, the bagged immature fruits accumulated a significant amount of anthocyanins after removing the bags. Combining RNA-seq analysis, exogenous ethylene treatment, and relative expression analysis, we confirmed that ethylene inhibits anthocyanin biosynthesis by repressing *SlAN2-like* expression in tomatoes. In addition, 86 transcription factors, including ERF, MYB, and bHLH, were identified from the transcriptome analysis and were highly correlated with the anthocyanin content. These findings provide new insights into anthocyanin regulation in purple tomato fruit.

## 4. Materials and Methods

### 4.1. Plant Materials and Treatments

The study used the purple–black tomato cultivator ‘Indigo Rose’. The plants were cultured in plastic greenhouses in Guangzhou, Guangdong province, China, 2021. After setting, the fruits were bagged with lightproof double-layered paper. The bags were removed when fruit developed at a certain stage in the experiments. Simultaneously, the surrounding leaves were removed to expose the fruits fully under natural light. The fruit peels were collected before (0 days, 0D) and at 2, 5, and 8 days after removing the bags and subjected to anthocyanin content measurement and qRT-PCR analysis.

For ethylene treatment, the immature fruits bagged for 35 days after bloom (DAB) were sprayed with 50 mg/L ethephon (ETH, an ethylene-releasing reagent) containing 0.05% Tween 20. Mock treated fruits were sprayed with distilled water containing 0.05% Tween 20. After treatment, the bags were removed. During the treatment, anthocyanin content and gene relative expression levels were measured at certain times. Finally, the fruits were peeled, frozen in liquid nitrogen immediately, and stored at −80 °C until use. All tests were conducted thrice.

### 4.2. Anthocyanin Extraction and Quantification

Tomato fruit peels collected from three fruits from different plants were used for anthocyanin quantification. Anthocyanin extraction and quantification were performed following previously published methods [53]. Briefly, the fruits were washed with tap water after sampling and then peeled. The peels were manually ground with liquid nitrogen into powder. 1 g of the powder was extracted with 10 mL HCl 1% (*v*/*v*) in methanol with the addition of two-thirds volume of distilled water. After a 12 h extraction at 4 °C, the samples were centrifuged at 3900 rpm for 20 min. The supernatant (2 mL) was filtered by 0.45 μm nylon membrane (Whatman 0.45 μm PVDF), and absorption was determined spectrophotometrically (A_535_ and A_650_). The amount of anthocyanin was determined spectrophotometrically (A_535_-A_650_) and expressed as mg of petunidin-3-(*p*-coumaroyl rutinoside)-5-glucoside per g, based on an extinction coefficient of 17,000 and a molecular weight of 934. All experiments were repeated three times.

### 4.3. Total RNA Isolation, cDNA Synthesis and Real-Time PCR Analysis

RNA extraction from frozen peels, cDNA synthesis, and real-time PCR was performed following previously described methods [42]. Briefly, total RNA was isolated using an Eastep Super RNA isolation kit (Promega, WI, USA). cDNA was synthesized from 1 ug of total RNA using the GoScript TM Reverse Transcription System (Promega, WI, USA). qRT-PCR was performed using previously described methods [41] using the tomato *ACTIN* (*Solyc03g078400*) gene as the reference. Next, all analyses were performed with three technical replicates. The 2^–^^△Ct^ method was used to calculate the relative gene expression [54]. The data were analyzed by one-way analysis of variance (ANOVA) using EXCEL 2016. The primers used for qRT-PCR are listed in Appendix A.

### 4.4. RNA-Seq Analysis

The fruit peels collected from the immature (35 DAB) and red ripening (48 DAB) stages before (0D) and after (2D) removing bags were used for RNA-seq analysis. The fruit peels were carefully split with a scalpel blade and rapidly frozen in liquid nitrogen. Five fruits from different plants were pooled to make a sample. Then, three biological replicates were used for each RNA sequencing treatment. The transcriptome libraries were generated using a NEBNext UltraTM RNA Library Prep Kit (New England Biolabs, MA, USA) designed for Illumina, following the manufacturer’s protocols. Indexed codes were added to attribute sequences to each sample. Next, the index-coded samples were clustered on a cBot Cluster Generation System using a HiSeq 4000 PE Cluster Kit (Illumina, CA, USA) following the manufacturer’s instructions. After cluster generation, the libraries were sequenced on an Illumina Hiseq 4000 platform (Illumina, CA, USA), generating 150 bp paired-end reads.

The raw reads were filtered using the Fastp software by removing the adaptor and low-quality sequences with the default parameters [55]. The STAR software was used to map the clean reads to the tomato reference genome (SL4.0) [56], and FeatureCounts was used to analyze the transcript levels of annotated genes [57]. Then, the transcript levels of each annotated gene (ITAG4.1) were counted and normalized as FPKM. The DEGseq package was used to calculate the *p*-values, and *p* < 0.01 was considered the threshold for identifying DEGs [58]. GO term analysis was conducted in the GENE ONTOLOGY (http://geneontology.org/ (accessed on 1 March 2022)) in biological processes using the most homologous genes in Arabidopsis [59].

## Figures and Tables

**Figure 1 ijms-23-07648-f001:**
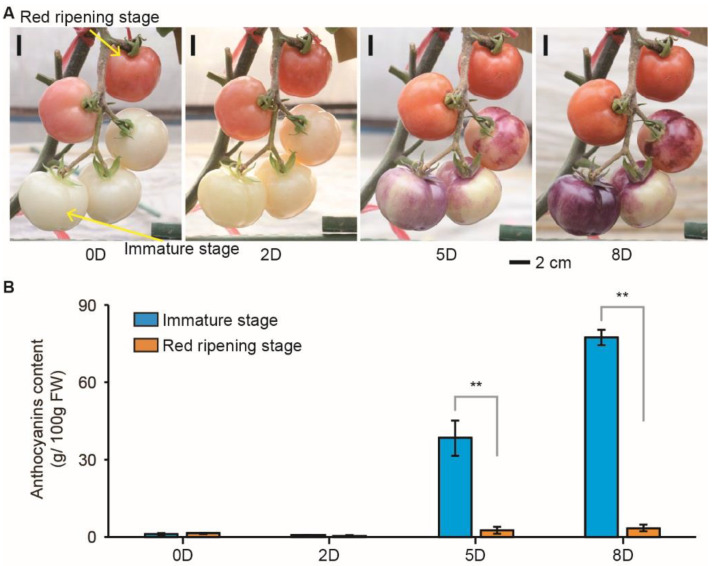
Fruit coloration and anthocyanin content analysis of immature and red ripening fruits of the purple tomato cultivar ‘Indigo Rose’ before and after removing bags. (**A**) The coloration of immature and red ripening fruits before (0D, 0 days) and after (2D, 5D, and 8D) removing bags. (**B**) Anthocyanin content in the fruit peel. Dates are presented as the mean ± SD of three biological replicates. The asterisks indicate significant differences, determined by the Student’s *t*-test (** *p* < 0.01); 0D, 2D, 5D, and 8D represent zero days before bag removal and 2, 5, and 8 days after removing bags.

**Figure 2 ijms-23-07648-f002:**
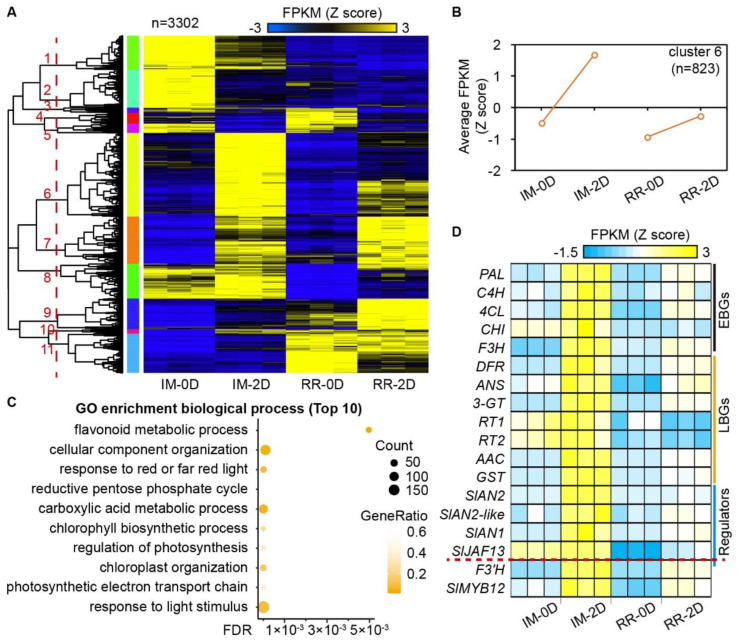
Global gene expression patterns in the immature and red ripening fruits before and after removing bags. (**A**) The expression profiles of 3302 differentially expressed genes (DEGs) in the immature and red ripening fruits before and after removing bags. The DEGs were clustered into 11 groups, indicated by the orange lines. (**B**) The expression profiles of genes in cluster 6. The y axis indicates the normalized FPKM (Z score). (**C**) The top ten enriched GO terms (biological process) by genes from cluster 6. (**D**) The expression of differentially expressed flavonoid-related genes between the immature and red ripening fruits. EBGs and LBGs mean early and late biosynthetic genes, respectively; IM—immature stage; RR—red ripening stage.

**Figure 3 ijms-23-07648-f003:**
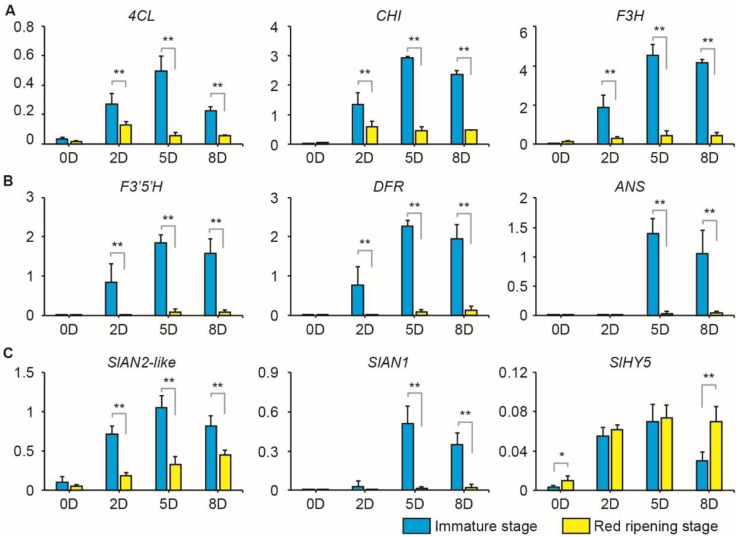
Transcriptional analysis of anthocyanin-related genes in immature and red ripening fruits before and after removing bags. Transcriptional analysis of the EBGs (**A**), LBGs (**B**), and transcription factors (**C**) using qRT-PCR. *ACTIN* was the reference gene. Data are presented as mean ± SD of three biological replicates. Asterisks indicate significant differences between immature and red ripening fruits, determined by the Student’s *t*-test (* *p* < 0.05, ** *p* < 0.01).

**Figure 4 ijms-23-07648-f004:**
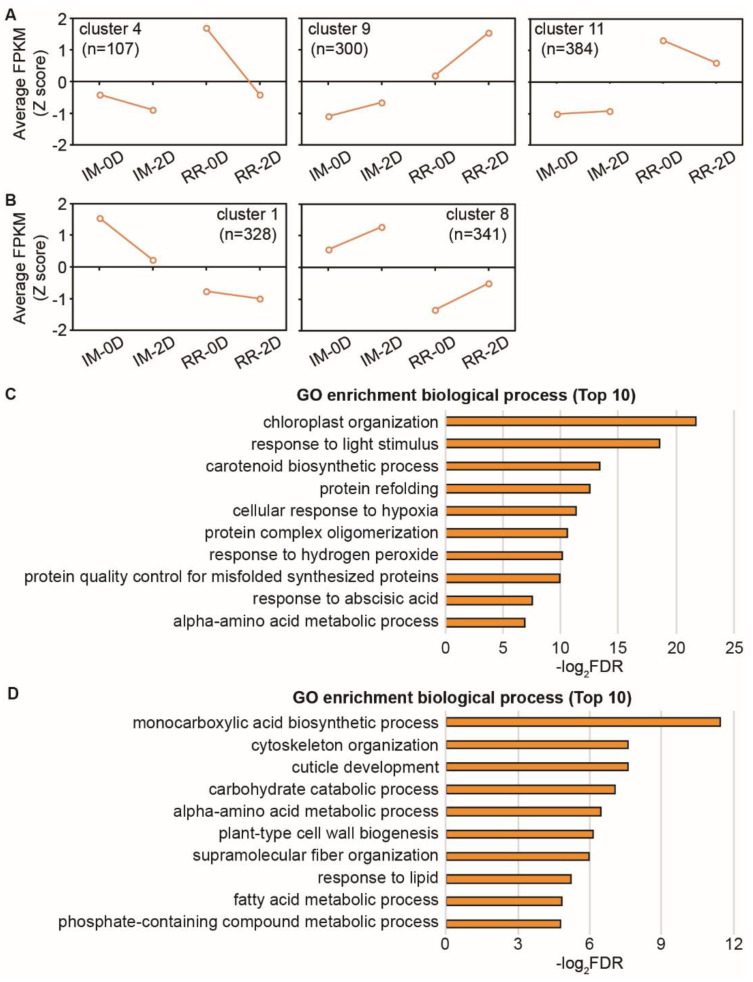
Co-expression analysis of the differentially expressed genes between immature and red ripening fruits. (**A**) The genes from clusters 4, 9, and 11 were more highly expressed in red ripening than in immature fruits. (**B**) The genes from clusters 1 and 8 had lower expression in red ripening than in immature fruits. (**C**) The top ten GO terms (biological processes) were enriched by genes from clusters 4, 9, and 11. (**D**) The top ten GO terms (biological processes) were enriched by genes of clusters 1 and 8.

**Figure 5 ijms-23-07648-f005:**
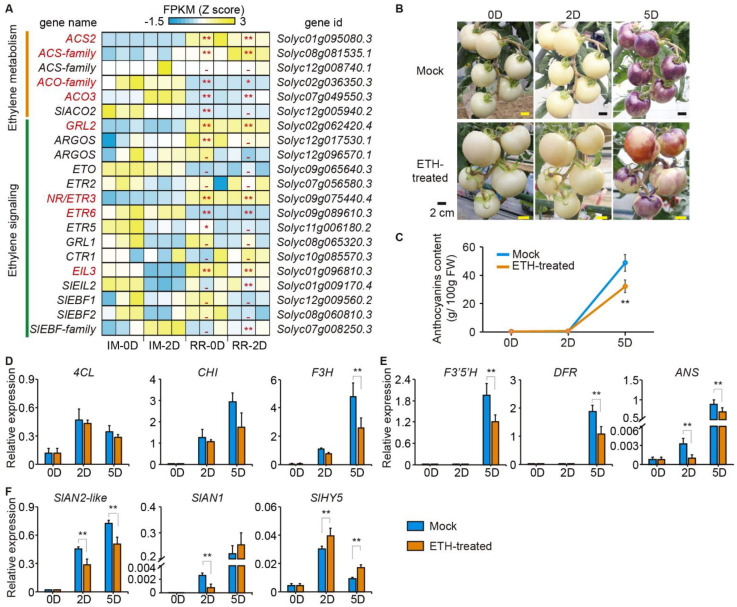
Effects of ethylene on anthocyanin accumulation and biosynthesis. (**A**) Expression profiles (FPKM, Z score) of the ethylene-related genes. Student’s *t*-test, n = 3; ** *p* < 0.01 and * *p* < 0.05 indicate statistically significant differences between immature and red ripening fruits. Fruit phenotype (**B**), anthocyanin content (**C**) and the relative expression levels of EBGs (**D**), LBGs (**E**), and regulators (**F**) in the ETH-treated and mock fruits before (OD) and after (2D and 5D) removing bags. Expression analysis was performed by qRT-PCR with *ACTIN* as the reference gene. Data are presented as the mean ± SD of three biological replicates. Asterisks indicate significant differences between immature and red ripening fruits, as determined by Student’s *t*-test (** *p* < 0.01). ETH, ethephon.

## Data Availability

Data are contained within the article or Appendix A.

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
