# Peer review of "Ethylene Inhibits Anthocyanin Biosynthesis by Repressing the R2R3-MYB Regulator SlAN2-like in Tomato"

_ijms, 2022, doi:10.3390/ijms23147648_

Round 1

Reviewer 1 Report

The present article is very interesting and it is well-written and, in my opinion, can open doors to more search. Excellent work! I only recommend a few things:

Line 32: Please, give some examples of biotic and abiotic factors.

Along with the article, and when convenient, change anthocyanin for anthocyanins, e.g., line 43 “Anthocyanins are synthesized”

Line 69: Tomato is a fruit, not a vegetable. Please, clarify this fact in the first sentence.

Line 76: 200 mg cyanidin 3-O-rutinoside /100 g, or what?

Line 175: bZit or BZIT, no?

I recommend making HPLC in order to enrich the paper and see what anthocyanins are found and at what levels.

Clarify 4.2.

Line 321: “scalpel blade”, no?

Reviewer 2 Report

In a reviewed manuscript, the authors investigated the influence of ethylene and fruit ripening on the biosynthesis of athocyanins in purple tomato fruit. The manuscript is based on transcriptomic data in conjunction with spectrophotometric determination of anthycyanins, which is the weak point of the entire manuscript. Quantification of individual anthocyanins would be desirable for a complete picture.

Overall, the experiment seems to be well performed and the results clearly presented.

L 304 Extraction and quantification of anthocyanins.
Although the authors used the published method, they need to briefly summarise the information on extraction and quantification, i.e., determination of the total amount of anthocyanins, because it is clear from the data that they did not quantify the individual anthocyanins. So you cannot use the word quantification of anthocyanins.

Also, I suggest adding a few sentences of conclusions at the end of the manuscript.
